# Water Thermodynamics and Its Effects on the Protein Stability and Activity

**Francesco Mallamace** [1,2,*] , **Domenico Mallamace** [3] , **Sow-Hsin Chen** [1], **Paola Lanzafame** [3] **and Georgia Papanikolaou** [3]

1   Department of Nuclear Science and Engineering, Massachusetts Institute of Technology, Cambridge, MA 02139, USA; sowhsin@mit.edu
2   Istituto dei Sistemi Complessi, Consiglio Nazionale delle Ricerche, 00185 Roma, Italy
3   Departments of ChiBioFarAm—Section of Industrial Chemistry, University of Messina, CASPE-INSTM, V.le F. Stagno d'Alcontres 31, 98166 Messina, Italy; mallamaced@unime.it (D.M.); paola.lanzafame@unime.it (P.L.); georgia.papanikolaou@unime.it (G.P.)
*   Correspondence: francesco.mallamace@unime.it

**Abstract:** We discuss a phenomenon regarding water that was until recently a subject of scientific interest: i.e., the dynamical crossover, from the fragile to strong glass forming material, for both bulk and protein hydration water. Such crossover is characterized by a temperature $T_L$ in which significant dynamical changes like the decoupling (or the violation of the Stokes-Einstein relation) of homologous transport parameters, e.g., the density relaxation time $\tau$ and the viscosity $\eta$, occur in the system. On this respect we considered the dynamic properties of water-protein systems. More precisely, we focused our study on proteins and their hydration water, as far as bulk and confined water. In order to clarify the effects of the water dynamical crossover on the protein properties we considered and discussed in a comparative way previous and new experimental data, obtained from different techniques and molecular dynamic simulation (MD). We pointed out the reasons for different dynamical findings from the use of different experimental techniques.

**Keywords:** water; protein folding; local order; relaxation times; self-diffusion; hydrophobic effect

## 1. Introduction

Water is one of the most exciting research topics; in particular, for its role in biology. It possesses one of the simplest molecular structures but in spite of this it is characterized, especially in the liquid state, by intriguing and complex thermodynamic properties which are far from completely understood [1–3]. Hence, if compared to other liquids, water has many anomalous behaviors, the best known example is the maximum density at 277 K. Nowadays we are certain that this complexity is due to its polymorphism [4], an original idea experimentally confirmed [5]. Like its amorphous phase [2], liquid water is also polymorphic; in particular, it is due to two liquids of different density: i.e., the high- and low-density liquids, HDL and LDL. The LDL structure is originated by a networking process with a tetrahedral symmetry due to the non-covalent attractive hydrogen bond interaction (HB). Relevant aspects are that HDL and LDL coexist in a large region of the water phase diagram and by changing pressure or temperature they can change one into the other by means of a first order transition: the liquid-liquid transition hypothesis (LLT) [4]. Polymorphism is reflected in the thermodynamic response functions (compressibility, specific heats, expansivity and entropy) of the system as well as in its variables, determining its full properties and fluctuations. This is valid in bulk water as well as in its solutions, proposing that for water, and water systems, the HB, reflected in the local molecular structure and configuration, determine their basic chemical-physical properties.

Proteins are intended in biology as the main functional macromolecules. Through temperature (or pressure) selections, they evolve in special functions depending on the

assumed structural configuration: the native-folded (*N*) and the unfolded-denatured state (*D*). Under certain thermodynamic limits their essential biological feature is the coherent (full-or-partial) way in which they switch forwards and backwards between these two basic states ($N \rightleftarrows D$) [6]. Such reversibility it is essentially based on three constraints: (i) proteins must fold in a reasonable time, (ii) the structure to which they fold must to perform a specific function, and (iii) the folded structure must be stable enough to perform it reliably. Above these limits, i.e., above at certain temperatures, the protein irreversibly denatures $N \rightarrow ID$.

As for water, the hydrophilic interaction, due to the HB, appears to be fundamental for the stability of the native phase of the protein as well as for its functionality and the reversibility of the ($N \rightleftarrows D$) process. The protein properties (in particular, folding-unfolding process) and the related physical effects of the water polymorphism represent a basic research topic today in science: very stimulating if we consider their correlations and the role of water as the "life's solvent" [7]. A compelling example of this is the pioneering discovery that proteins cannot perform their function if they are not covered by a minimum amount of water: the minimum or first hydration layer [8,9].

In this paper, we discuss the role of water, and the HB interactions, in the protein folding $\rightleftarrows$ unfolding thermally activated process by considering the results of recent experiments (like: light and neutron scattering and Nuclear Magnetic Resonance spectroscopy).

Protein hydration is the process for which the incrementally water addition, to a dry protein, restores its biological properties. Beyond the first hydration layer, further water addition is simply a dilution [8,9]. This shell is defined as the water associated with the protein at the hydration end point and it constitutes a monolayer covering the protein surface. Water outside this monolayer weakly interacts with the protein. Therefore, the macromolecule has two different kind of interactions with water: the bound internal water and the hydration water on its surface. Both interactions are believed to play an important role in controlling the protein biofunctionality. It was demonstrated, by measuring the reaction of lysozyme with the hexasaccharide of Nacetylglucosamine, that the enzymatic activity is closely parallel to the development of surface motion, which is thus responsible for the functionality of the protein [9].

The bound internal water molecules, in the protein cavities, are involved in local HB which, depending on the temperature (or pressure), can be sufficiently strong to maintain the protein in its folded globular phase. In this situation the HB must counteract, and overcome, the opposite effects of the hydrophobic interactions and electrically charged molecular groups otherwise the protein unfolds. This contrast between hydrophilicity and hydrophobicity is known as the hydrophobic effect (HE) and it seems to depend on both temperature and pressure. Recent NMR experiments showed a singular behavior of HE with *T*: for temperatures lower than the ambient one, the hydrophilicity dominates (the strength of which progressively increases as *T* decreases), while at higher temperatures the hydrophobicity essentially governs the system properties. The idea that hydration water plays a role in the protein stability was proposed for the first time by Kauzmann just by considering the "hydrophobicity" concept [10]. In addition, the conformational flexibility of a protein and therefore its functionality are extremely sensitive to the characteristics of its HBs with hydration water.

A second process that marks the protein properties, characterized by a sharp change in the protein mean square displacement $\langle x^2 \rangle$ (MSD) and observed at about $T_C = 220$ K, is the so-called protein glass transition. Below it the protein is in a state with solid-like structure executing harmonic vibrations and hardly shows any biological function [11,12]. As *T* increases atomic motions evolve from such state of harmonic solid to anharmonic liquid-like motions (above $T_C$) [13].

That transition in the protein dynamic, as proved by experiments [12,14] and simulations [15] is solvent induced. Such a result points out to a dynamic coupling between the solvent with the internal protein motions, suggesting that the cooperativity of the HB solvent network provides both the coupling mechanism and the protein stability. Fourier

transform infrared spectroscopy (FTIR) experiments give coherent results with such ideas proving that the hydration water crossover is the result of a transition from predominantly low density form of water at lower temperature (the LDL a less fluid state) to predominantly high density form at higher temperature (HDL a more fluid state) [16]. This finding is fully agree with Quasi-Elastic Neutron Scattering [17,18] and a MD simulation experiments on the same system [19] for which this hydration water crossover is the result, due to a progressive temperature increase, of a transition from predominantly LDL (lower $T$) to predominantly HDL water (higher $T$).

The proteins unfolding denaturation ($R$) takes place, increasing $T$, by means of intermediate structures [6]. This is a dynamic process determined by the energetic landscape (EL) typical of complex systems or supercooled glass forming materials (and supercooled water in particular) [20,21]. In fact, for the folding kinetic hypothesis structural intermediates, between the $R$ and $N$ states, are unstable and contribute to an energy barrier separating the native globular protein from the unfolded denatured. In the EL frame, the protein native structure corresponds to a relatively deep local minimum [22], whereas in the thermodynamic hypothesis such a minimum is global [23]. A great amount of conformational sub-states in the protein EL can be therefore expected. The highest energy barrier is defined as the protein folding transition state (FTS) and despite the high dimensionality of the folding reaction, it behaves like a transition state for a simple low-dimensional reaction with a simple exponential time evolution. The process equilibrium constant, in the fully reversible process $N \rightleftarrows R$ is described by the of rate constant for folding ($k_f$) and unfolding ($k_u$) as: $K_{R-N} = [R]/[N] = k_u/k_f$ in a fully reversible process $N \rightleftarrows R$, thus giving a simple formalism.

Another example of folding model, which expressly refers to the basic role of water, is the sequential one based on the concepts of hydrophilicity and hydrophobicity. According to it, folding takes place via these stages: (i) the formation in an unfolded chain of a secondary structure element, stabilized by peptide-HB, (ii) the merging of pre-existing blocks with a secondary structure to another intermediate globular, stabilized by the hydrophobicity, and (iii) the final stabilization of this latter to the native structure HB and "van der Waals interactions". The internal water molecules are thus involved in all of these states and can act, especially in the native globular phase, as local bridges among protein hydrophilic parts to stabilize its folded structure.

Taking into consideration what has been said, this work is aimed to a further verification of these strong correlations between water and the protein folding process. In this frame we have also considered the relevance for water and water systems of a singular and thermodynamically consistent temperature, $T^* = 315 \pm 5$ K, that reasonably is the origin of its anomalies [24]. In the $P - T$ water phase diagram, at such temperature the water isothermal compressibility, $K_T(P, T)$, shows a minimum for all the studied pressures and all the corresponding coefficient of thermal expansion $\alpha_P(P, T)$ curves cross in a "singular and universal point" $\alpha_P(T^*) \cong 0.44 \ 10^{-3}$ K$^{-1}$). Furthermore, from the structural point of view, $T^*$ may be considered as the onset temperature of the HB tetrahedral clustering (or of the LDL water phase), the "magic point" at which liquid water crosses from a simple normal liquid to an anomalous complex material. This situation is essentially related to the HB lifetime. In fact, near $T^*$ the HB is of the order of the picosecond [25] and therefore unable to sustain the HB network (and the LDL phase) as well as insufficient to maintain the protein in the folded globular configuration. In contrast, by cooling it increases exponentially on following the water networking.

In particular, we used dynamic and structural data measured in hydrated lysozyme. The lysozyme shows intermediate structures under chemical [26], pressure-induced [27] and thermal denaturation [28,29]. As proposed by the calorimetric measurements [28,29], its unfolding process can be considered as a three-state model: native state ($N$) $\rightleftarrows$ intermediate (unfolded and possibly reversible, $R$ ) state $\longrightarrow$ denatured (irreversibly $ID$ ) state. The first step is represented by the native state followed by the reversible denaturation and can be also considered as a kind of strong-to-fragile liquid transition associated with

the configurational entropy change [29,30], while the final one is the irreversible denaturation (*ID*) and it is due to an association of unfolded lysozyme units [27,31]. All these experimental findings were confirmed by theory [32] and simulation [33].

Therefore, taking into account data from previous experiments [16,17,19,34,35] and considering a special analysis performed at molecular level by means of the NMR technique, we discuss the protein thermal denaturation and the water role in terms of the HB interaction.

## 2. The State of the Art

As said the hydrated lysozyme was studied through different techniques. Before proceeding further, it is appropriate to show how NMR well describes, by means of the water self-diffusion data $D_S(T)$, both the protein glass transition and the unfolding process, and also provides appropriate information on their correlation at molecular level with water. The Figure 1 illustrates in an Arrhenius plot the measured $D_S(T)$ vs. $1000/T$ for the lysozyme at three different hydration levels: $h = 0.3, 0.37$ and $0.48$, together with those of pure liquid water. The studied temperature range is, for water, $180 < T < 373$, whereas for hydrated lysozyme $200 < T < 367$.

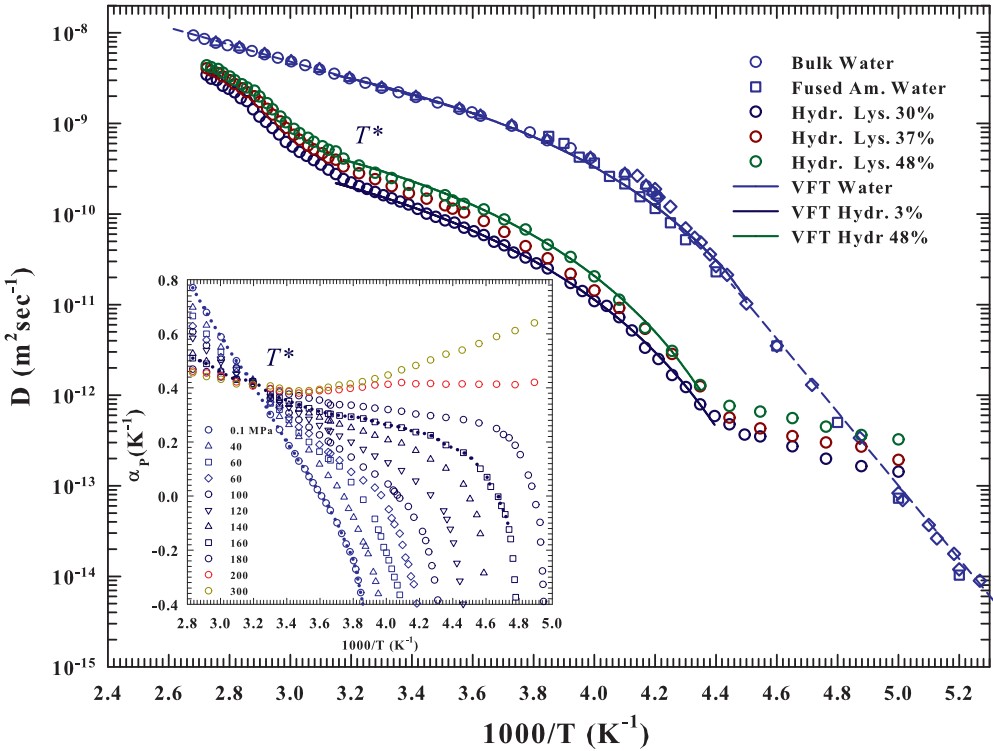

**Figure 1.** The Arrhenius plot ($lnD_S$ vs. $1000/T$) of the proton self-diffusion $D_S(T)$ measured in the lysozyme at three different hydration levels: $h = 0.3, 0.37$ and $0.48$, together with those of pure liquid water. The studied temperature range is, for water, $180 < T < 373$, wheras for hydrated lysozyme $200 < T < 367$. The inset reports the pure bulk water expansion coefficient $\alpha_P$ in the $P, T$ plane. The corresponding data behaviors well illustrate the thermodynamic singularity of the temperature $T^*$.

The pure water data [36–40] show two-dynamical crossover: the first Arrhenius to super-Arrhenius at $T^* = 315 \pm 5$ K ($1000/T^* \simeq 3.18$ K$^{-1}$) and the second one, super-Arrhenius to Arrhenius (straight lines), in the very deep supercooled regime at about $T_L = 223$ K. As can be observed the super-Arrhenius region between these two characteristic temperatures is well described by the Vogel-Fulcher-Tammann (VFT equation—continuous curve). While the first crossover at $T^*$ was associated, as said, with the LDL limit, the second is instead a universal characteristic of the dynamic arrest of supercooled glass forming liquids (the locus of onset of the so-called hydrodynamic heterogeneities

where the Stokes-Einstein law is violated). A data fitting inside the two crossovers by using the VTF ($D = D_0 exp(-BT_0/T - T_0)$) give $T_0 = 183$ K, $B = 1.77$ and $D_0 = 3.45 \times 10^{-8}$ m$^2$s$^{-1}$, and below $T_L$ the Arrhenus law ($D = D_0 \exp -E_A/RT$) $E_A = 21.2$ kcalmol$^{-1}$. Whereas [36], for $T > T^*$ it is $E_A = 3.2$ kcalmol$^{-1}$. The hydrated protein shows, for the three different hydration levels, and within the experimental error, the two crossovers at about the same water, along with almost identical super-Arrhenius behavior (always inside the crossovers temperature). In fact, we measured for all the $h$ levels that $D_0$ increases with $h$ from $2.94 \times 10^{-9}$ to $5.26 \times 10^{-9}$m$^2$s$^{-1}$, whereas the other two parameter have values very close to those of the solvent ($185 < T_0 < 191$ K and $1.71 < B < 1.74$. Outside the two crossovers ($T < T_L$) the behaviors remain Arrhenius but with a different activation energy $E_A \simeq 3.48$ kcalmol$^{-1}$ (neutron experiments measure a value of $E_A \simeq 3.13$ kcalmol$^{-1}$) [17,41].

For $T > T^*$ the protein-water $D_S$ presents a sigmoid behavior almost identical for all $h$ but completely different from the Arrhenius showed by the bulk water and therefore due to the unfolding process, and as proposed by the calorimetry data associated with the water-protein interactions. Such behavior can be clarified on considering the water properties evidenced by the behavior of its expansivity $\alpha_P$ in the $P, T$ plane and illustrated in the inset of the Figure 1. As it is well known such important thermodynamic function reflect the behavior of the $V - S$ cross-correlations, being $\alpha_P = \frac{1}{V}(\partial V/\partial T)_P = \langle \delta S \delta V \rangle / k_B T V$. The corresponding isobars of water show together with the impressive "singular and universal point"at $T^*$ all the singular behaviors characterizing its liquid phase if compared with normal fluids and due to the basic polymorphism. Such anomalies, as evidenced by this plot, starting from $T^*$ (inside the stable phase) characterize all the supercooled regime in all the pressure range $0.1 < P < 200$ MPa. Its important highlight that the fluctuations $\delta S$ and $\delta V$, on cooling decrease, whereas in water they are pronounced. Furthermore, in regular liquids are always positively correlated and few pressure dependent. In addition, in water, below $T_m$ (where it is $\alpha_P < 0$), they become anti-correlated (an increase in $V$ brings an $S$ decrease). The pronounced entropy decrease, by decreasing $T$, from the stable liquid to inside the supercooled state, meaning that the water cooling is accompanied by a marked increase in its local order. This is a further evidence of the onset and development of the molecular clustering driven by the HB: the LDL phase.

FTIR experimental data of the OH stretching vibrational spectra (OHS) [42] and MD [43] simulation findings fully confirmed this reality by showing that the LDL phase begins to form just near $T^*$ after which, decreasing $T$, it increases (while HDL decreases and becomes the dominant phase at $T_L$). The same analysis made in a FTIR experiment on the lysozyme hydration water in the stable liquid range ($273 < T < 373$ K) [16], and reported in the inset of the Figure 2, gives further confirmation of these considerations. In this context the presence of an isosbestic point in both Raman and FTIR OHS spectra constitutes a strong evidence of a "mixture model" of water involving HB and non-hydrogen bonded (NHB) molecules [44]; the LDL phase is that of the tetrahedral network whereas the HDL and LDL consists of the contribution HB (trimers and dimers) and NHB (monomers). The same picture emerges from a MD analysis [45], where the distribution of tetrahedralities is likewise bimodal. The Figure 2 reports the measured specific heat as a function of $T$, for the water/lysozyme system ordinarily obtained in the study of the unfolding process [28], and gives an overall agreement, on molecular terms, between the specific heat data and that of the FTIR OHS vibrational spectra results. The protein unfolding process, accompanied by an early stage of reversibility starts just when the population of HDL molecules is at the maximum. And its ID process starts just when the population of NHB molecules approaches that of the HB ones, i.e., just when the probability for water molecules to form a HB and a not-bonded structure is about the same.

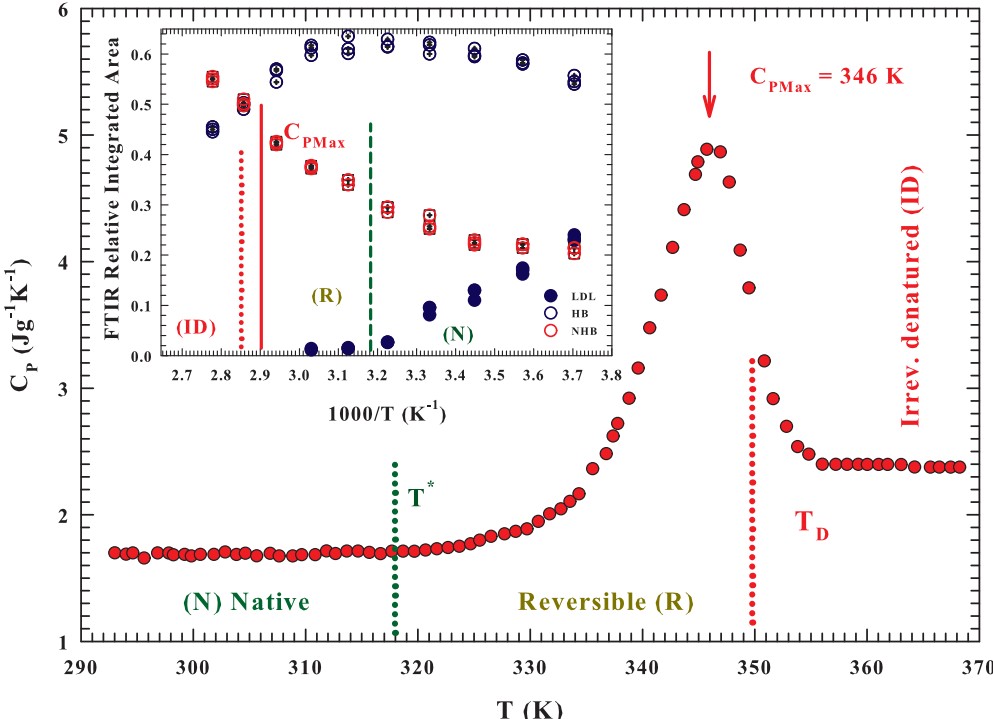

**Figure 2.** The measured specific heat evolution, as a function of $T$, in the thermal range $290 < T < 370$ K, for the water/lysozyme system originally obtained in the study of the unfolding process [28]. The inset gives, in an Arrhenius plot (range $270 < T < 357$ K) the FTIR OHS vibrational contributions of the hydrated lysozyme ($h = 0.3, 0.37$ and $0.48$). In particular, the contributions of hydrogen bonded (HB and LDL) and non-hydrogen bonded (NHB) molecules are considered according to the "mixture model" [44]. The HDL water phase is made of HB and NHB.

As said, these calorimetric studies reveal, on following the scheme: $(N) \rightleftarrows (R) \longrightarrow (ID)$ for the hydrated lysozyme, that the thermal denaturation involves an intermediate reaction [28]. The measured $C_p(T)$ broad peak, on considering some heating-cooling cycles, is ascribed to thermally reversible contributes. The data similarity suggested that there is a reversible equilibrium between the native state $N$ and a conformationally distinguishable intermediate state $R$; and that the $N \longrightarrow ID$ in an endothermic transformation. As $T$ is increased, the equilibrium constant $K_{R-N}$ increases (according to the vant-Hoff equation) and the population of state $R$ increases as the population of state $N$ decreases. Finally, state $R$ irreversibly converts to the more denatured state $ID$ with a rate constant $k$. Thus, the $N$ state does not directly denature but, prior the denaturation, it undergoes a rapid conformational change, and all the unfolded protein states are in $R$. The maximum $C_{p,\max}$ is observed at $346.2 \pm 1$ K.

The measured vanishingly small values of both the $R$ state weight fraction and $k$, showed that the fractional amount of $D$ is negligible at relatively low temperatures. As the hydrated protein is furtherly heated, $R$ and $k$ increases. During the early part of heating the formed $D$ fraction remains however negligible, when $k$ is still small. Hence, $C_p$ is the sum of the corresponding values of both the $N$, and $R$ states, with the contributions from the thermal effects owing to the reversible and rapid $N \rightleftarrows R$ transformation.

This calorimetric study also shows that some significant temperature intervals can be easily identified: (i) $T \leq 318$ K, where the $C_p$ is due to the vibrational and configurational degrees of freedom of protein native state $N$; (ii) $318 > T > 346$ K, characterized by a relatively rapid increase indicating a predominant conversion of state $N$ to state $R$ ($K_{R-N}$ increases with $T$ and thus the amount of state $R$, at equilibrium increases; (iii) 346 to 356 K, where the rapid $C_p$ decrease is a result of a relatively slow increase in the amount of the intermediate state; whereas a relatively rapid increase take place in the post-denaturation.

For $T > 356$ K, the slight heat capacity change is caused by the vibrational and configurational contributions of the denatured lysozyme, and from any further denaturation. The lysozyme reversibility region stops at the end of the $C_{p,\max}$ region, i.e., near $T_D = 356$ K.

Both the Figure 2 and its inset show the characteristic temperatures of the three-state model $N \rightleftarrows R \longrightarrow ID$: $T^*$, $T_D$ and also the maximum specific heat ($C_{PMax}$).

This is the current situation regarding the lysozyme folding-unfolding process coming out from experiments and confirmed by MD simulations. We presume that the role of water and in particular the HB between water-protein and water-water is at the basis of this fundamental process. We believe that protein unfolding process starts just when the HB strength (and life time) is not enough to keep the protein in its native state, and that in this process the water singular temperature $T^*$ plays a special role as its molecular trigger. To clarify and underline this intriguing reality we considered other properties measurable with NMR and Neutron spectroscopies.

## 3. Methods.

### 3.1. Hydrated Protein Preparation

Egg white lysozyme used in this experiment was obtained from Fluka (L7651 three times crystallized, dialyzed, and lyophilized) and used without further purification. Samples were dried, hydrated isopiestically, and controlled by means of a precise procedure [17] The sample was lyophilized to remove any water. The dried protein powder was then hydrated isopiestically at 278 K by exposing it to water vapor in a closed chamber until hydration level ($h = 0.30, 0.37$ and $0.48 \pm 0.01$ is reached i.e., $0.35, 0.432$ and $0.56$ gram of water per gram of dry lysozyme, respectively. The hydration level was determined by thermogravimetric analysis and also confirmed by directly measuring the weight of the absorbed water. This hydration level $h = 0.30$ corresponds to about a monolayer coverage on the protein surfaces. For each experimental run we used different samples and the temperature was maintained with an accuracy of $\pm 0.2$ K.

### 3.2. NMR Experiments

The dynamic properties of the system lysozyme-hydration water was studied at ambient pressure and different temperatures by using a Bruker AVANCE NMR spectrometer operating at 700 MHz $^1H$ resonance frequency. To explore all the $N \rightleftarrows R \longrightarrow ID$ process, the hydrated protein was heated from 295 to 365 K (which steps of 2 K, slowly made in $\sim$20 min avoiding abrupt temperature variation). In these experiments, we studied: (a) the longitudinal proton relaxation times $T_1$ (or spin-lattice) of the water protons and (b) the $^1H$-NMR spectra (obtained from the free-induction decay (FID)) by measuring: (i) the proton chemical shift (($\delta$) as the shift of the central frequency value, by using the methanol $\delta(T)$ as a $T$ standard), (ii) the maximum intensity $I^{\max}$ and (iii) the apparent spin-spin relaxation time $T_2^*$ by the HWHM peak width, $\Delta \nu$ as $T_2^* = 1/(\pi 2 \Delta \nu)$. $T_2^*$ is the rate of the so-called apparent spin–spin relaxation, which is related to the spin–spin relaxation time-constant $T_2$. (a quantity related to the proton rotational time, i.e., a measure of the interparticle orientational time). The spectroscopic experimental configuration was in the "Magic Angle Spinning (MAS)". By tilting samples of a precise angle (about 54.7°) with respect to the direction of the applied magnetic field, the Hamiltonian term corresponding to dipolar interactions vanishes and NMR peaks become narrower [46]. Hydrated protein powder were placed in a 50 μL rotor and spun at 4000 Hz at the magic angle to increase the spectral resolution.

The $T_1$ was measured by using the standard inversion recovery pulse sequence ($\pi - t - \pi/2$ acquisition, with $t$ denoting the time between the two rf pulses). In contrast to the pure bulk water whose measured spectra are represented by a single exponential time decay, the protein hydration spectra instead contain two $T_1$ contributions; one due to the surface water and the second one to the protein protons. Furthermore, being the NMR signal intensity $I(T)$ directly related with the equilibrium magnetization of the studied material, $M_0$ (or the susceptivity $\chi_0$) (linearly dependent on the total number of mobile

spins per unit volume, on the mean square value of nuclear magnetic moment and on $1/T$ (Curie law)) the measured spectra were corrected for the Curie effect.

## 4. Discussion

First of all, the $T^*$ role can be explained, on microscopic basis, by considering a neutron scattering results; whereas the Adam–Gibbs model taking into account the slope changes in the Arrhenius plot of the transport functions confirms that specific heat maxima are essentially due to configurational effects [47]. We will first consider the temperature evolution of the molecular migration distance $d$, reported in the upper panel of the Figure 3 in an Arrhenius plot. While the self-diffusion constant represents how fast a molecule diffuses, $d(T)$ represents, in the frame of the Singwi–Sjölander model of water trapped in a cage formed by adjacent molecules connected by HBs [48], how far the center of mass of a typical molecule translates. Such a quantity can be obtained from a Quasi-Elastic-Neutron-Experiment (QENS) from the measured incoherent dynamic structure factor $S_H(Q, E)$ (translational contribution), in the wave-vector limit $Q \to 0$ [35]. One can see, from Figure 3, that $d(T)$ is increasing slowly below $T^*$ from 4.2 to 4.6 Å but rises sharply above to 9.6 Å at 380 K. Indicating a large scale enhanced movement of the water molecules when the lifetime of the HB network of the water molecules becomes shorter, and thus it is not able to maintain the shape of the protein (above $T^*$).

The Adam–Gibbs equation allows to calculate the configurational entropy $S_{Conf}$ from the diffusion coefficient $D$ [47]:

$$D(T) = D_0 \exp(-A/T S_{Conf}) \tag{1}$$

being $D_0$ is a prefactor, that can be estimated from the $D(T)$ in the high $T$ limit, and $A$ is a constant. Thus, from $S_{conf}$ can be obtained $C_{P,conf} = T(\partial S_{conf}/\partial T)_P$. If we assume for $A$ −31.75 kJmol$^{-1}$ the value suggested by a simulation study [49], $S_{Conf}$ and $C_{P,conf}$ can be calculated. The Figure 3 bottom panel proposes the obtained result in the range $200 < T < 370$ K, and as can be seen, two maxima can be observed, corresponding respectively to the protein glass transition ($\sim$225 K) and to the unfolding process ($\sim$346 K). Furthermore, between the two peaks, a decreasing and linear behavior can be observed in the configurational specific heat, ending just near $T^*$, where $C_{P,conf}$ has a minimum. A behavior this latter, in fully agreement with the $d(T)$ behavior, that again highlights how the liquid water HB (and polymorphism) is determinant both for its thermodynamic properties and the protein stability and behaviors. At this minimum, and therefore at $T^*$, the HB networking (and the LDL phase) becomes unstable, due to the increasing weakness of HB on the one hand and the HE effect (accompanied by the energy landscape, strongly temperature dependent) the on the other, so that further temperature increases determine the onset of the unfolding process, the reversible transition ($N \rightleftarrows R$) first, and after the $ID$ process. Hence, above this specific temperature the thermal fluctuations (energy or Brownian motion) prevail leading to HBs braking.

The Figure 4 illustrates, in an Arrhenius plot, both the Magnetization ($M_0$ upper panel) and the apparent spin-spin relaxation time ($T_2^*$ bottom) measured for $h = 0.3$ by using two different samples. Both these quantities show extremes at $T^*$ (maximum) and at the temperature of the $C_{p,\max}$. In particular, by increasing $T$, from the protein native state, both these quantities increase until the state of reversibility unfolding is reached, after which a decrease begins, which ends where there is the maximum of configurational variations leading to irreversible unfolding.

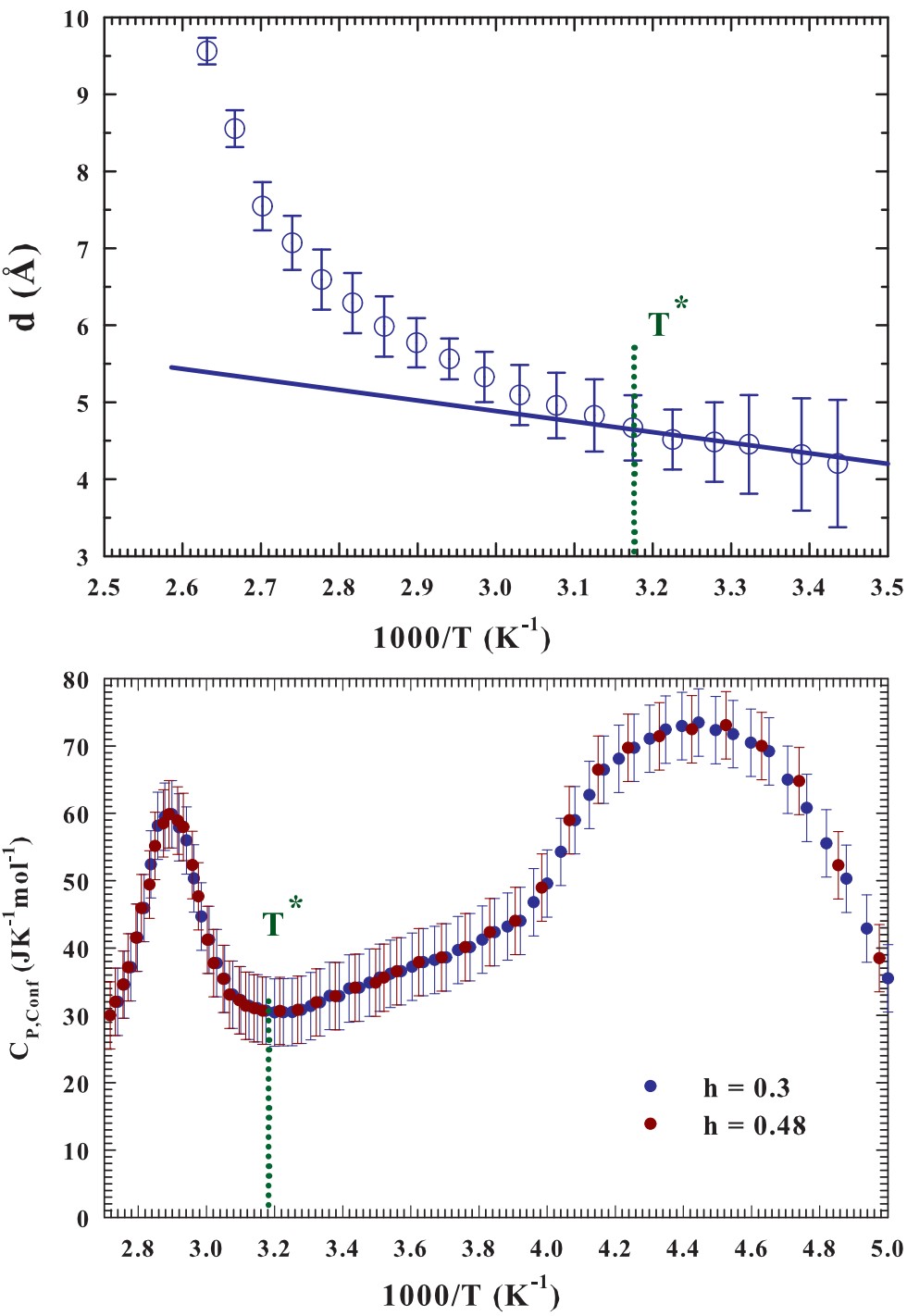

**Figure 3.** The upper panel illustrates in an Arrhenius plot the temperature evolution of the molecular migration distance *d*, calculated from neutron scattering data measured in the interval 290–380 K [35]. Whereas the bottom one proposes, in a similar plot, the configurational specific heat. $C_{P,conf}$ evaluated according to the Adam-Gibbs model [47] (range 200 < *T* < 370 K) from the lysozyme hydration water self-diffusion $D_S$.

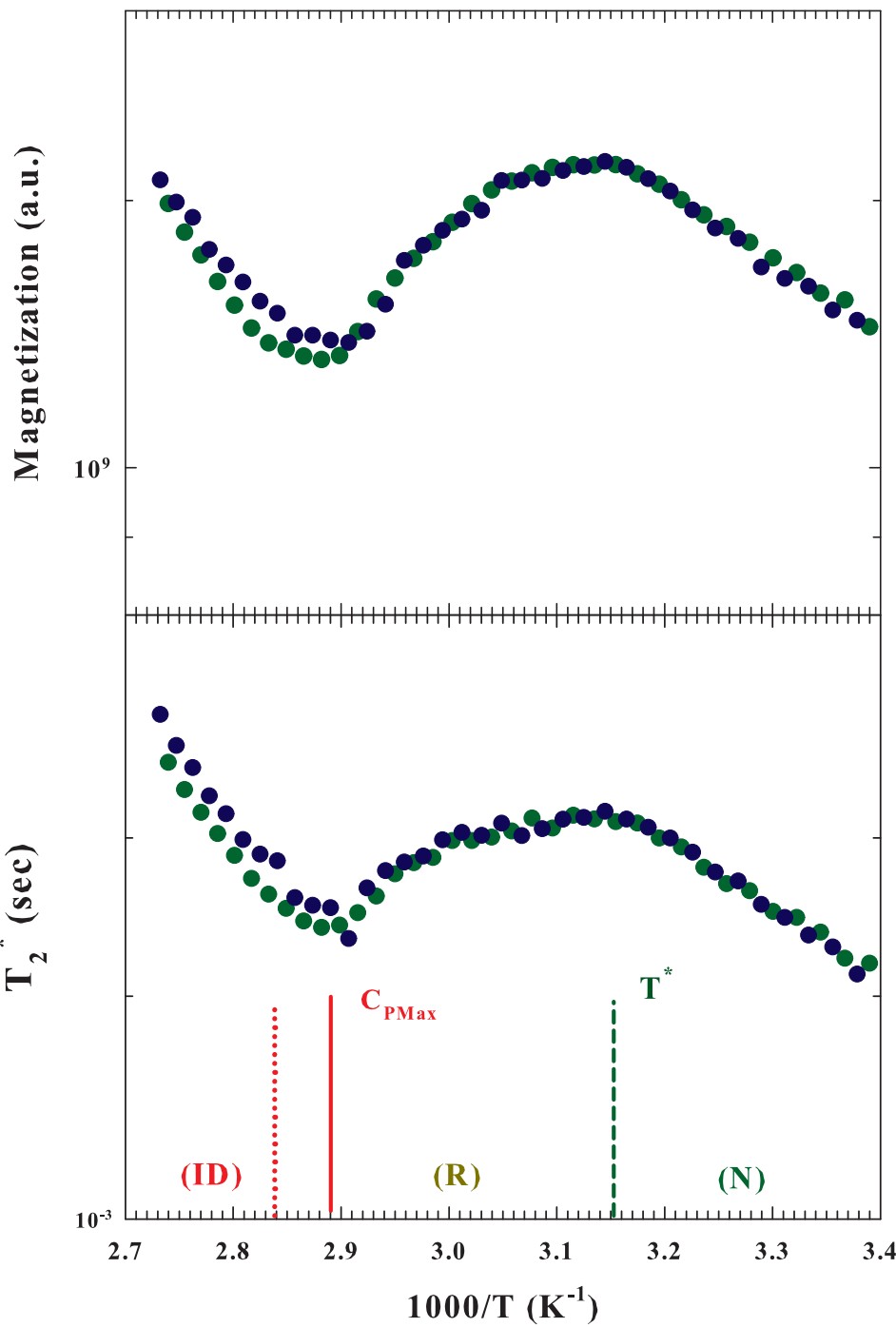

**Figure 4.** The Arrhenius plot, of the Magnetization ($M_0$ upper panel) and the apparent spin-spin relaxation time ($T_2^*$ bottom) measured for $h = 0.3$ by using two different samples. For both the quantities the temperature range is the same (295–366 K). $M_0$ evolves from 1.6 to $1.16 \cdot 10^{-9}$ (a.u.), whereas $T_2^*$ from 2.5 to $1.5 \cdot 10^{-3}$ (sec).

In general, relaxations measured in an NMR experiment are caused by random fluctuations of the magnetic field at the position of a resonating spin originating from the thermal motion of neighboring spins [50]. The fluctuating magnetic dipole–dipole interactions between $^1H$ spins are thus due to the tumbling of molecules under the local caging structure. Hence, the observed behaviors of both $M_0$ and $T_2^*$ are related to the evolutions in the water local structure, and in particular to its packing density, and the interactions with the

protein. A situation that reflects all interaction effects in the unfolding process, in both the system structure and dynamics measured via the $^1H$ relaxations.

As said the protein hydration NMR spectra show two $T_1$ contributions, one due to the hydration water and the second one to the protein protons. The Figure 5 (top) illustrates the proton relaxation rates ($R_1 = 1/T_1$) corresponding to these contributions (measured at $h = 0.30$) and that of the pure bulk water; in the figure are shown the two characteristic temperatures corresponding to the $N \rightleftarrows R \longrightarrow ID$ process together with that of the $C_{p,\max}$. As can be observed, the spin-lattice relaxation time of the protein internal water is faster than that of the hydration and pure water, and shows an opposite behavior as $T$ increases. $R_1$, according to the current theoretical models [50–52] is related to the reorientational correlations, which the corresponding time in water increases (like $R_1$) by decreasing temperature [53]. The bottom side of the Figure 5 reports the relative weights of these two contribution measured in the hydrated protein and how it can be observed, by increasing $T$, these quantities remain constant up to $T^*$. After which the contribution of the hydrating water increases to the detriment of the internal one to the protein which tends to disappear suggesting that the protein irreversible folding is accompanied by a protein dehydration due to contrast between the HB and the HE at these temperatures. This situation is due to the behavior of the corresponding proton relaxation rates (Figure 5, top side). Also in this case the $R_1$ corresponding protein internal water remains nearly constant inside the native state after which approaching the ID grows by almost an order of magnitude. Looking instead at the contributions of hydration and bulk water, it can be noted (beyond a difference in their values of half an order of magnitude) an identical thermal behavior (super-Arrhenius) inside the native state. Instead when, increasing $T$, the $R_1$ of pure water crosses to the Arrhenius behavior, that of the hydration water begins to grow slightly up to $C_{p,\max}$, after which it decreases tending to the values of bulk water inside the *ID* region. All this once again confirms the key role of water-protein interactions in the protein denaturation process. Furthermore the evident super Arrhenius behavior of these hydration water data provides proof that proteins unfolding denaturation ($R$) is a dynamic process determined by the energetic landscape (EL) [22].

Finally, we focused on the measured chemical shift behavior of the protein hydration water from the native state to the ID. $\delta$ is the linear response of the system electronic structure to an external magnetic field [54,55]. It is related to the magnetic shielding tensor $\sigma$, that dependent on the local electronic environment, and is a useful probe for the system local geometry; in particular, for the HB structure of water, aqueous systems and solutions [56]. Of interest are its isotropic part, $\sigma_{\text{iso}} \equiv \text{Tr}(\sigma/3)$ and anisotropy $\Delta\sigma$; $\sigma_{\text{iso}}$ is experimentally obtained via the measured $\delta$ relative to a reference state [51], because the deviation of $\sigma(T)$ from a reference value gives $\delta(T)$. Since the magnetic susceptibility per water molecule, $\chi_0$, can be assumed to be $T$ and $P$ independent, an isolated water molecule in a dilute gas can be taken to be the reference and $\delta$ represents the effect of the interaction of water with the surroundings providing, in particular, a rigorous picture of the intermolecular geometry [57], being directly related to the average number of HBs in which a water molecule is involved $\langle N_{HB} \rangle$, ( $\delta(T) \propto \langle N_{HB} \rangle$) [57–59]. Hence, $\langle N_{HB} \rangle$ represents the number of possible water configurations, so that it can be assumed: $S_{Conf} \approx -k_B \ln \langle N_{HB} \rangle$, and the $\delta(T)$ derivative $-(\partial\delta(T)/\partial T)_P \approx -(\partial \ln \langle N_{HB} \rangle / \partial T)_P \approx (\partial S_{Conf}/\partial T)_P$ should be proportional to the configurational contribution of the constant pressure specific heat.

The Figure 6 illustrates the thermal evolution of the measured $^1H$ NMR chemical shift $\delta(T)$ during the temperature induced protein unfolding of the hydrated lysozyme at $h = 0.3$, along with that of pure water (top). As can be observed while the chemical shift of pure water shows a decreasing and linear behavior as $T$ increases, that of hydration water is characterized by discontinuity (a kink), within two linear regions, that occurs inside the reversible unfolding phase on approaching the *ID* region (from about 339 to 350 K). This behavior reflects the strong change in the system structure just due to the protein unfolding process. The resulting configurational specific heat evaluated as: $C_{P,conf} = -T(\partial\delta(T)/\partial T)_P$ [34] is shown in the figure bottom. As can be seen, from

a comparison of this latter result with the specific heat measured (Figure 2) and that evaluated by means of the Adam-Gibbs (Figure 3 bottom), the resulting agreement is certainly significant.

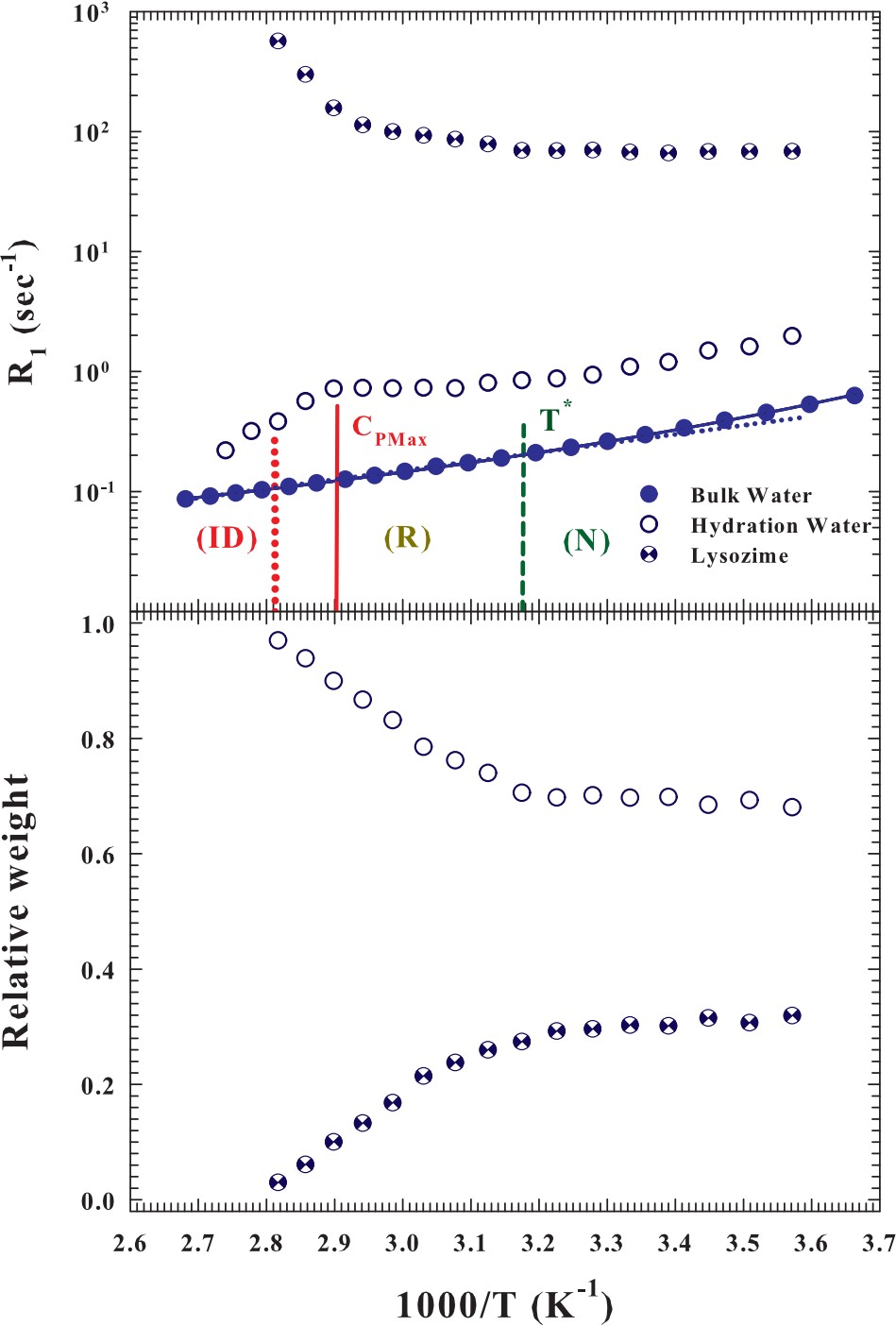

**Figure 5.** The figure bottom side proposes the Arrhenius plot of the proton relaxation rates ($R_1 = 1/T_1$, measured in the lysozyme at $h = 0.30$) for the hydration and internal protein water and that of the pure bulk water; the two characteristic temperature corresponding to the $N \rightleftarrows R \longrightarrow ID$ process together with that of the $C_{p,\max}$ are also shown. Instead the bottom side of the figure illustrates the relative weight of the two protein water contributions.

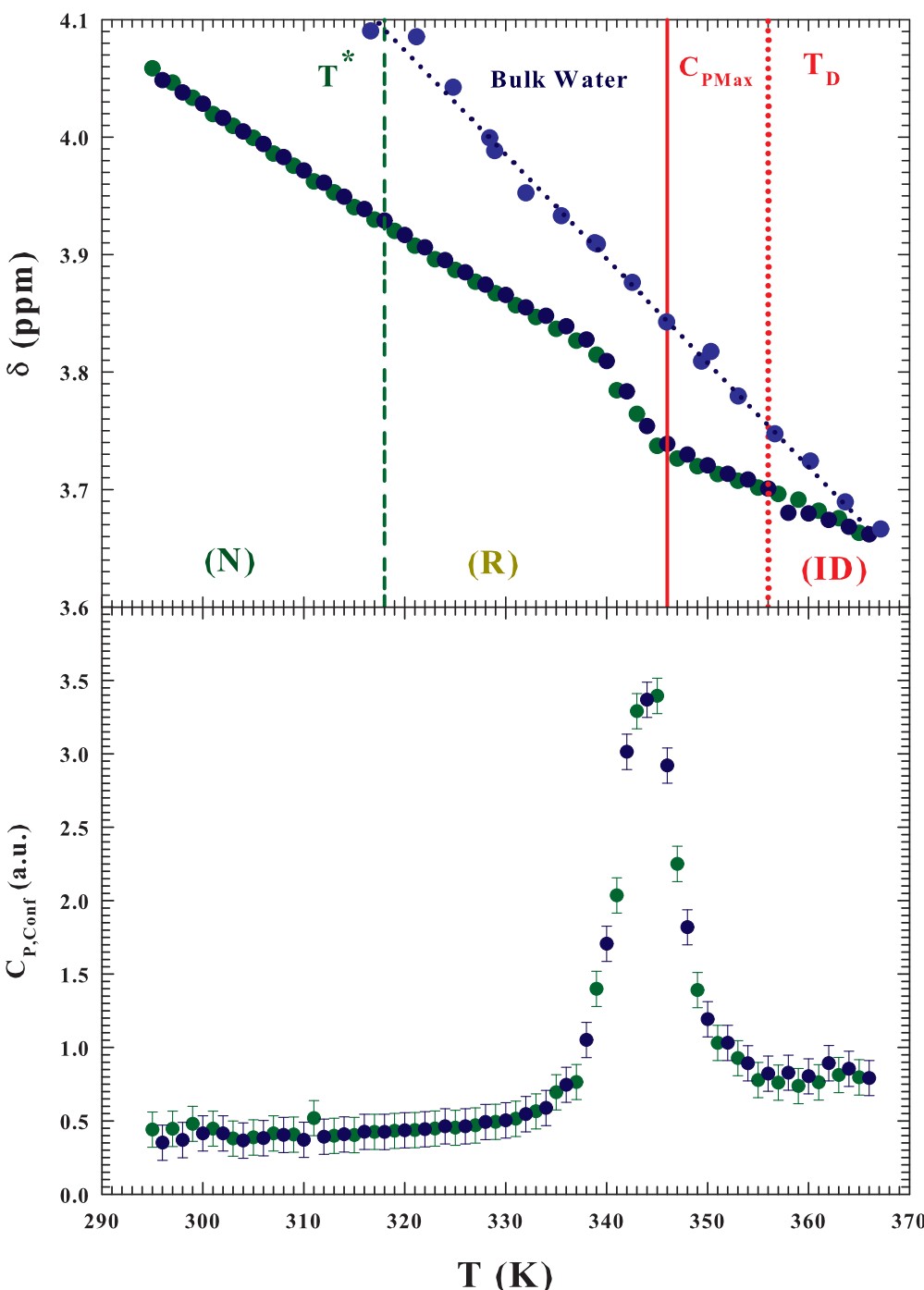

**Figure 6.** Top panel: the thermal evolution, in the range 295–368 K, of the measured $^1H$ NMR chemical shift $\delta(T)$ during the temperature induced protein unfolding of the hydrated lysozyme at $h = 0.3$, along with that of pure water. Instead the bottom of the figure proposes the corresponding configurational specific evaluated, by considering that the chemical shift is a measure of the atomic local order, as: $C_{P,conf} = -T(\partial\delta(T)/\partial T)_P$ [34].

## 5. Concluding Remarks

In this work we considered the thermal folding of lysozyme in water at different hydration levels and the related experimental findings coming from different experimental techniques. Our main interest was directed in order to clarify the role, in this complex phenomenon, of the water molecules around and inside the protein. We have specifically considered the polymorphism of this liquid characterized by complex thermodynamic

anomalies and both the role of the HB interaction and that of the hydrophobic effect. Following the suggestions of a previous calorimetric study [28] in which was proved that the thermal protein denaturation take places via a three stage model. i.e., starting from the evolving of the native structure (compact and globular) to an intermediate (globular, open or molten) through a reversible transition, and finally it leads to the irreversible change of this latter in an essentially unfolded polymer chain (a disordered coil). In this conformational evolution from the native globular structure to that of an open coil the internal protein interaction is switched off and the macromolecular packing decreased at each of the steps characterizing the $N \rightleftarrows R \longrightarrow ID$ process. In the firm belief of a central role of water in all of this, based on the HB interaction (that in some thermodynamic conditions is a powerful cage forming mechanism) we considered an accurate analysis of the $^1H$ NMR data to probe the water dynamic and structure during the entire protein unfolding.

The obtained results, also combined with neutron scattering and FTIR data not only confirmed that water as a local probe follows accurately all the proteins changes in the thermal denaturation of the lysozyme but also evidenced that the HB, plays a determinant role in the process.

We first examined the hydration and bulk water self-diffusion coefficient in a very large temperature range including both the protein glass transition and protein denaturation process. These $D_S(T)$ data gave evidence of a crossover just at the temperature where the calorimetry data show that the onset of the first stage ($N \rightleftarrows R$) of the protein unfolding is located in the immediate vicinity of $T^*$, i.e., at the temperature where, by increasing $T$, as shown by: (i) the FTIR OHS data, (ii) those of the molecular migration distance $d$, and (iii) the singular behavior of the water thermal expansion $\alpha_P(P,T)$ of the water molecular clustering and thus the water polymorphism (driven by the HB) is no longer energetically supported. A thermodynamic situation in which the LDL phase is minimal (or fully disappeared) and only the HDL phase (HB + NHB) remains. Above $T^*$ where the HB population has its maximum, after which that of the NHB grows and just when the behaviors of the two populations cross, the definitive irreversible unfolding takes place. Subsequent increases in temperature lead to the predominance of NHB-free water over HB and therefore to the disappearance of the water-protein interaction. Conditions in which the macromolecule becomes definitively a disordered few interacting coil: a polyelectrolyte dissolved in a solvent.

After which we considered the data provided by the NMR technique and essentially focused essentially on the proton spectra: the intensity (the molecular magnetization $M_0$) and the anomalous spin-spin relaxation time ($T_2^*(T)$). Both these quantities evidence that protein water not only follow all the three stages of the denaturation process but also reveals the determinant role of water. Again through NMR we measured (at $h = 0.30$) the proton spin-lattice relaxation time ($T_1$), and then analyzed the corresponding relaxation rates $R_1 = 1/T_1$. In this case was possible to separately monitor the thermal evolution of the contribution due to hydration water and to that internal to the protein; as well as their relative weights. Also in this last case the evolution with the temperature of the data and a comparison with those corresponding to the bulk water showed that in the protein reversible unfolding the water-protein interactions play a key role, also originating the conditions for the energetic landscape (EL).

The system configurational specific heat $C_{P,conf}$, was obtained by using the Adam-Gibbs model through the measured proton self-diffusion data. Such a quantity is characterized by two maxima (one at the temperature of the protein glass transition and the other at the temperature as the maximum the unfolding process) and near $T^*$. This result fully confirms that the water molecules follow the changes in the protein structure. Finally the proton chemical $\delta(T)$ and its derivative $-(\partial\delta(T)/\partial T)_P$ related respectively with the local order and again to the configurational specific heat confirm as water and its HB interaction determine the entire thermal evolution of the hydrated protein and thus its

biological activity and functionality, these properties being strongly linked to the unfolding process [6].

**Author Contributions:** All authors contributed equally. All authors have read and agreed to the published version of the manuscript.

**Funding:** This research received no external funding.

**Data Availability Statement:** The data that support the findings of this study are available from the corresponding author upon reasonable request.

**Acknowledgments:** The D.M. work was supported by the European Project H2020 A-LEAF-732840; L.P. and G.P. benefited from the national PRIN 2017 project (Italy).

**Conflicts of Interest:** The authors declare no conflict of interest.

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
