# Peer review of "Water Thermodynamics and Its Effects on the Protein Stability and Activity"

_biophysica, doi:10.3390/biophysica1040030_

Round 1

Reviewer 1 Report

In attachement.

Author Response

First of all, we want to thank both referees for the very useful suggestions that have allowed us to improve the manuscript. In essence, we have carefully followed what they have pointed out, including careful manuscript editing.

In particular, all the changes made are reported in bold.

Regarding the referee 1 main comment “how the strength (energy) and duration of HBs can be concluded from NMR data”. We referred to the main NRM literature, reported in the work references 50-56, which shows how an accurate analysis of the two main NMR relaxation times (T1 and T2) can provide a molecular measure of both these quantities (or quantities directly related to them). A situation also linked to the evolution in temperature and pressure of self-diffusion. In other words, these works (especially that of Bloembergen (50) and that of Kubo (52)) fortunately show how the NMR relaxations accurately reflect the system local fluctuations and their effects.

Technical issues:

  • Is word “controversy” usen 2 times in abstract the right choice? Does somebody disagree with the presence of this crossover? Who and why? If Authors present such statement in abstract it should be explained later in the text. We have eliminated this word.
  • About terminology: Authors are using different forms of “Hydrogen-bond interaction” or “hydrogen bonding HP interactions” etc. Maybe just “hydrogen bond” would be enough? Abbreviation HB is defined many times in different forms. Done in accordance with the suggestion.
  • Please provide FTIR abbreviation explanation. Done in accordance with the suggestion.
  • Page 5 – N,R,ID abbreviations are provided but they are not explained until page 8 – please change the order or add (N),(R), (ID) on the page 5 below the process scheme. Done in accordance with the suggestion.
  • Page 7, third line “17?” – unnecessary question mark. It was an incorrect citation, that we corrected.

Reviewer 2 Report

In the manuscript entitled “Water thermodynamics and its effects on the protein stability and activity”, the authors addressed a very divisive topic. A series of complementary experiments were used to demonstrate the role of both bulk and hydration water on lysozyme stability. I would like to highlight the relevance of this issue, and the effort of the authors on extensively prove the effects of the hydration water on protein stability.

The experiments are well described, the paper is well written and the data shown is significant and warrant publication. Therefore, my recommendation is to accept the manuscript after minor revision (small text editing/minor spell check required, and perhaps changes in the title so that is not that much general).

Author Response

First of all, we want to thank both referees for the very useful suggestions that have allowed us to improve the manuscript. In essence, we have carefully followed what they have pointed out, including careful manuscript editing.

In particular, all the changes made are reported in bold.

Reply to the referee 2 main comment: minor revision (small text editing/minor spell check required, and perhaps changes in the title so that is not that much general).        The work has been carefully edited, but if possible we would like to keep the current title

Round 2

Reviewer 1 Report

The work still needs to be polished and in parts rewritten by a native English speaker - it was suggested in a previous review. Below are some examples of errors in a use of English language (chosen randomly from the paper):

  • Page 7, “In this context must be stressed that…”
  • Page 8, “Hence, Cp is the sum of the corresponding to the state N, and R, and the measured value contains contributions from any thermal effects owing to the reversible and rapid N _ R transformation.
  • Page 10 “To measure of T1 was used the standard inversion recovery pulse sequence”
  • Page 11 “In starting with our discussion we consider a neutron scattering result”
  • From the Adam-Gibbs equation can be calculated the configurational entropy SConf from the diffusion coefficient D48:
  • Page 11: “By using the Adam-Gibbs model we evaluat from the measured proton self-diffusion data the corresponding system configurational specific heat CP;conf , a quantity characterized by two maxima (one at the temperature of the protein glass transition and the other at the same temperature as the specific heat maximum characterizing the unfolding process) and a minimum in the T_ vicinity.”
  • Page 13 “due to the HB smallness”
  • “A situation the latter better defined by the behavior of the corresponding proton relaxation rates”
  • Page 14 “All of this is reflects the strong change in the system structure”

Author Response

Thanks again for your suggestions. In according with you the paper was revised by an English native.